# Learning Attractor Dynamics
# for Generative Memory

**Yan Wu, Greg Wayne, Karol Gregor, Timothy Lillicrap**
DeepMind
{yanwu,gregwayne,karolg,countzero}@google.com

## Abstract

A central challenge faced by memory systems is the robust retrieval of a stored pattern in the presence of interference due to other stored patterns and noise. A theoretically well-founded solution to robust retrieval is given by attractor dynamics, which iteratively clean up patterns during recall. However, incorporating attractor dynamics into modern deep learning systems poses difficulties: attractor basins are characterised by vanishing gradients, which are known to make training neural networks difficult. In this work, we avoid the vanishing gradient problem by training a generative distributed memory without simulating the attractor dynamics. Based on the idea of memory writing as inference, as proposed in the Kanerva Machine, we show that a likelihood-based Lyapunov function emerges from maximising the variational lower-bound of a generative memory. Experiments shows it converges to correct patterns upon iterative retrieval and achieves competitive performance as both a memory model and a generative model.

## 1   Introduction

Memory plays an important role in both artificial and biological learning systems [4]. Various forms of external memory have been used to augment neural networks [5, 14, 25, 29, 31, 32]. Most of these approaches use attention-based reading mechanisms that compute a weighted average of memory contents. These mechanisms typically retrieve items in a single step and are fixed after training. While external-memory offers the potential of quickly adapting to new data after training, it is unclear whether these previously proposed attention-based mechanisms can fully exploit this potential. For example, when inputs are corrupted by noise that is unseen during training, are such one-step attention processes always optimal?

In contrast, experimental and theoretical studies of neural systems suggest memory retrieval is a dynamic and iterative process: memories are retrieved through a potentially varying period of time, rather than a single step, during which information can be continuously integrated [3, 7, 20]. In particular, attractor dynamics are hypothesised to support the robust performance of various forms of memory via their self-stabilising property [8, 12, 16, 28, 33]. For example, point attractors eventually converge to a set of fixed points even from noisy initial states. Memories stored at such fixed points can thus be retrieved robustly. To our knowledge, only the Kanerva Machine (KM) incorporates iterative reconstruction of a retrieved pattern within a modern deep learning model, but it does not have any guarantee of convergence [32].

Incorporating attractor dynamics into modern neural networks is not straightforward. Although recurrent neural networks can in principle learn any dynamics, they face the problem of *vanishing gradients*. This problem is aggravated when directly training for attractor dynamics, which by definition imply vanishing gradients [23] (see also Section 2.2). In this work, we avoid vanishing gradients by constructing our model to dynamically optimise a variational lower-bound. After training, the stored patterns serve as attractive fixed-points to which even random patterns will

converge. Thanks to the underlying probabilistic model, we do not need to simulate the attractor dynamics during training, thus avoiding the vanishing gradient problem. We applied our approach to a generative distributed memory. In this context we focus on demonstrating high capacity and robustness, though the framework may be used for any other memory model with a well-defined likelihood.

To confirm that the emerging attractor dynamics help memory retrieval, we experiment with the Omniglot dataset [22] and images from DMLab [6], showing that the attractor dynamics consistently improve images corrupted by noise unseen during training, as well as low-quality prior samples. The improvement of sampling quality tracks the decrease of an energy which we defined based on the variational lower-bound.

## 2 Background and Notation

All vectors are assumed to be column vectors. Samples from a dataset $\mathcal{D}$, as well as other variables, are indexed with the subscript $t$ when the temporal order is specified. We use the short-hand subscript $_{<t}$ and $_{\leqslant t}$ to indicate all elements with indexes "less than" and "less than or equally to" $t$, respectively. $\langle f(\mathbf{x}) \rangle_{p(\mathbf{x})}$ is used to denotes the expectation of function $f(\mathbf{x})$ over the distribution $p(\mathbf{x})$.

### 2.1 Kanerva Machines

Our model shares the same essential structure as the Kanerva Machine (figure 1, left) [32], which views memory as a global latent variable in a generative model. Underlying the inference process is the assumption of *exchangeability* of the observations: i.e., an episode of observations $\mathbf{x}_1, \mathbf{x}_2, \ldots, \mathbf{x}_T$ is exchangeable if shuffling the indices within the episode does not affect its probability [2]. This ensures that a pattern $\mathbf{x}_t$ can be retrieved regardless of the order it was stored in the memory — there is no forgetting of earlier patterns. Formally, exchangeability implies all the patterns in an episode are conditionally independent: $p(\mathbf{x}_1, \mathbf{x}_2, \ldots \mathbf{x}_T | \mathbf{M}) = \prod_{t=1}^{T} p(\mathbf{x}_t | \mathbf{M})$.

More specifically, $p(\mathbf{M}; \mathbf{R}, \mathbf{U})$ defines the distribution over the $K \times C$ memory matrix $\mathbf{M}$, where $K$ is the number of rows and $C$ is the code size used by the memory. The statistical structure of the memory is summarised in its mean and covariance through parameters $\mathbf{R}$ and $\mathbf{U}$. Intuitively, while the mean provides materials for the memory to synthesise observations, the covariance coordinates memory reads and writes. $\mathbf{R}$ is the mean matrix of $\mathbf{M}$, which has same $K \times C$ shape. $\mathbf{M}$'s columns are independent, with the same variance for all elements in a given row. The covariance between rows of $\mathbf{M}$ is encoded in the $K \times K$ covariance matrix $\mathbf{U}$. The vectorised form of $\mathbf{M}$ has the multivariate distribution $p(\mathrm{vec}(\mathbf{M})) = \mathcal{N}(\mathrm{vec}(\mathbf{M})| \mathrm{vec}(\mathbf{R}), \mathbf{I} \otimes \mathbf{U})$, where $\mathrm{vec}(\cdot)$ is the vectorisation operator and $\otimes$ denotes the Kronecker product. Equivalently, the memory can be summarised as the matrix variate normal distribution $p(\mathbf{M}) = \mathcal{MN}(\mathbf{R}, \mathbf{U}, \mathbf{I})$, Reading from memory is achieved via a weighted sum over rows of $\mathbf{M}$, weighted by addressing weights $\mathbf{w}$:

$$\mathbf{z} = \sum_{k=1}^{K} \mathbf{w}(k) \cdot \mathbf{M}(k) + \xi \tag{1}$$

where $k$ indexes the elements of $\mathbf{w}$ and the rows of $\mathbf{M}$. $\xi$ is observation noise with fixed variance to ensure the model's likelihood is well defined (Appendix A). The memory interfaces with data inputs $\mathbf{x}$ via neural network encoders and decoders.

Since the memory is a linear Gaussian model, its posterior distribution $p(\mathbf{M}|\mathbf{z}_{\leqslant t}, \mathbf{w}_{\leqslant t})$ is analytically tractable and online Bayesian inference can be performed efficiently. [32] interpreted inferring the posterior of memory as a writing process that optimally balances previously stored patterns and new patterns. To infer $\mathbf{w}$, however, the KM uses an amortised inference model $q(\mathbf{w}|\mathbf{x})$, similar to the encoder of a variational autoencoder (VAE) [19, 27], which does not access the memory. Although it can distil information about the memory into its parameters during training, such parameterised information cannot easily by adapted to test-time data. This can damage performance during testing, for example, when the memory is loaded with different numbers of patterns, as we shall demonstrated in experiments.

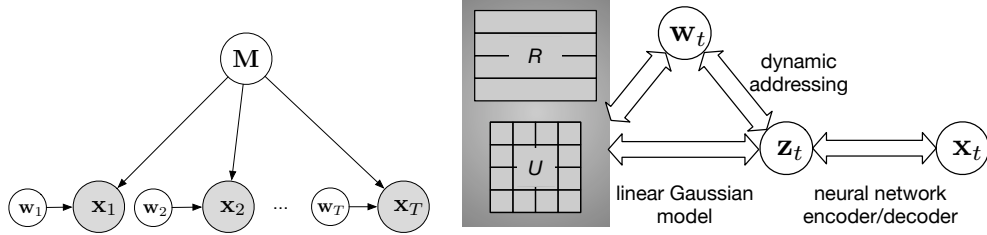

Figure 1: Variables: $\mathbf{M}$ – the memory, $\mathbf{x}$ – inputs (e.g., images), $\mathbf{w}$ – addressing weigths, $\mathbf{z}$ – embedding of $\mathbf{x}$. Left: The probabilistic graphical model shared by the Kanerva Machine and our model. The memory is a latent variable shared by all patterns in an episode, and provides exchangeability within the episode. $\mathbf{z}$ is omitted since the deterministic embedding of $\mathbf{x}$ does not affect the graphical model. Right: Schematic structure of our model. The memory is a Gaussian random matrix.

## 2.2 Attractor Dynamics

A theoretically well-founded approach for robust memory retrieval is to employ attractor dynamics [3, 15, 18, 33]. In this paper, we focus on point attractors, although other types of attractor may also support memory systems [12]. For a discrete-time dynamical system with state $\mathbf{x}$ and dynamics specified by the function $f(\cdot)$, its states evolve as: $\mathbf{x}_{n+1} = f(\mathbf{x}_n)$. A fixed-point $\mathbf{x}^* = \mathbf{x}_n$ satisfies the condition $\frac{\partial \mathbf{x}_{n+1}}{\partial \mathbf{x}_n} = \frac{\partial f(\mathbf{x})}{\partial \mathbf{x}}\big|_{\mathbf{x}=\mathbf{x}_n} = 0$, so that $\mathbf{x}_{n+1} = \mathbf{x}_n = \mathbf{x}^*$. A fixed point $\mathbf{x}^*$ is *attractive* if, for any point near $\mathbf{x}^*$, iterative application of $f(\cdot)$ converges to $\mathbf{x}^*$. A more formal definition of a point attractor is given in Appendix E, along with a proof of attractor dynamics for our model. Gradient-based training of attractors with parametrised models $f(\cdot; \theta)$, such as neural networks, is difficult: for any loss function $\mathcal{L}$ that depends on the $n$'th state $\mathbf{x}_n$, the gradient

$$\frac{\partial \mathcal{L}}{\partial \theta} = \frac{\partial \mathcal{L}}{\partial \mathbf{x}_n} \cdot \sum_{t=1}^{n} \frac{\partial \mathbf{x}_n}{\partial \mathbf{x}_t} \cdot \frac{\partial \mathbf{x}_t}{\partial \theta} \qquad (2)$$

vanishes when $\mathbf{x}_t$ approaches a fixed point, since $\frac{\mathbf{x}_n}{\mathbf{x}_t} = \prod_{u=t}^{n-1} \frac{\mathbf{x}_{u+1}}{\mathbf{x}_u} \to \mathbf{0}$ when $\mathbf{x}_u \to \mathbf{x}^*$ according to the fixed-point condition. This is the "vanishing gradients" problem, which makes backpropagating gradients through the attractor settling dynamics difficult [23, 24].

## 3 Dynamic Kanerva Machines

We call our model the Dynamic Kanerva Machine (DKM), because it optimises weights $\mathbf{w}$ at each step via dynamic addressing. We depart from both Kanerva's original sparse distributed memory [18] and the KM by removing the static addresses that are fixed after training. The DKM is illustrated in figure 1 (right). Following the KM, we use a Gaussian random matrix $\mathbf{M}$ for the memory, and approximate samples of $\mathbf{M}$ using its mean $\mathbf{R}$. We use subscripts $t$ for $\mathbf{M}_t$, $\mathbf{R}_t$ and $\mathbf{U}_t$ to distinguish the memory or parameters after the online update at the $t$'th step when necessary. Therefore, $p(\mathbf{M}_t | \mathbf{x}_{\leqslant t}) = p(\mathbf{M} | \mathbf{x}_{\leqslant t})$.

We use a neural network encoder $e(\mathbf{x}) \to \mathbf{z}$ to deterministically map an external input $\mathbf{x}$ to embedding $\mathbf{z}$. To obtain a valid likelihood function, the decoder is a parametrised distribution $d(\mathbf{z}) \to p(\mathbf{x}|\mathbf{z})$ that transforms an embedding $\mathbf{z}$ to a distribution in the input space, similar to the decoder in the VAE. Together the pair forms an autoencoder.

Similar to eq. 1, we construct $\mathbf{z}$ from the memory and addressing weights via $\mathbf{z} = \sum_{k=1}^{K} \mathbf{w}(k) \cdot \mathbf{M}(k)$. Since both mappings $e(\cdot)$ and $d(\cdot)$ are deterministic, we hereafter omit all dependencies of distributions on $\mathbf{z}$ for brevity. For a Bayesian treatment of the addressing weights, we assume they have the Gaussian prior $p(\mathbf{w}_t) = \mathcal{N}(\mathbf{0}, \mathbf{1})$. The posterior distribution $q(\mathbf{w}_t) = \mathcal{N}(\mu_{w_t}, \sigma_w^2)$ has a variance that is trained as a parameter and a mean that is optimised analytically at each step (Section 3.1). All parameters of the model and their initialisations are summarised in Appendix B.

To train the model in a maximum-likelihood setting, we update the model parameters to maximise the log-likelihood of episodes $\mathbf{x}_{\leqslant T}$ sampled from the training set (summarised in Algorithm 1). As

---

**Algorithm 1** Training the Dynamic Kanerva Machine (Single training step)

sample an episode $\mathbf{x}_1, \mathbf{x}_2, \ldots, \mathbf{x}_T$ from $\mathcal{D}$
Initialise memory $q(\mathbf{M}_0) = p(\mathbf{M}_0; \mathbf{R}_0, \mathbf{U}_0)$
**for** t = 1 : T (in arbitrary order) **do**                                          // begin writing
   compute embedding $\mathbf{z}_t = e(\mathbf{x}_t)$
   compute weights distribution $q(\mathbf{w}_t)$ by solving $\mu_{\mathbf{w}_t}$ from eq. 6 using $q(\mathbf{M}_{t-1})$
   update memory (Appendix A): $q(\mathbf{M}_t; \mathbf{R}_t, \mathbf{U}_t) \leftarrow q(\mathbf{M}_{t-1}|\mu_{\mathbf{w}_t}, \mathbf{z}_t; \mathbf{R}_{t-1}, \mathbf{U}_{t-1})$
   (optional) set $q(\mathbf{M}_{t-1}) = q(\mathbf{M}_t)$ and repeat the previous 2 steps
**end for**                                                                            // end of writing
**for** t = 1 : T (in arbitrary order) **do**                                          // begin reading
   compute embedding $\mathbf{z}_t = e(\mathbf{x}_t)$
   compute weights distribution $q(\mathbf{w}_t)$ by solving $\mu_{\mathbf{w}_t}$ from eq. 6 using $q(\mathbf{M}_T)$
   compute read-out embedding: $\hat{\mathbf{z}}_t \leftarrow \sum_{k=1}^{K} \mathbf{w}_t(k) \cdot \mathbf{R}_T(k)$ using sample $\mathbf{w}_t \sim q(\mathbf{w}_t)$
**end for**                                                                            // end of reading
compute the the objective $\mathcal{O} \leftarrow \mathcal{L}_T + \mathcal{L}_{\text{AE}}$ (eq. 4 and eq. 11) using previously obtained terms
update parameters via gradient ascent to maximise $\mathcal{O}$

---

is common for latent variable models, we achieve this by maximising a variational lower-bound of the likelihood. To avoid cluttered notation we assume all training episodes have the same length $T$; nothing in our algorithm depends on this assumption. Given an approximated memory distribution $q(\mathbf{M})$, the log-likelihood of an episode can be decomposed as (see full derivation in Appendix C):

$$\ln p(\mathbf{x}_{\leqslant T}) = \mathcal{L}_T + \sum_{t=1}^{T} \langle \mathrm{D}_{\mathrm{KL}}(q(\mathbf{w}_t) \| p(\mathbf{w}_t | \mathbf{x}_t, \mathbf{M})) \rangle_{q(\mathbf{M})} + \mathrm{D}_{\mathrm{KL}}(q(\mathbf{M}) \| p(\mathbf{M} | \mathbf{x}_{\leqslant T})) \quad (3)$$

with its variational lower-bound:

$$\mathcal{L}_T = \sum_{t=1}^{T} \left( \langle \ln p(\mathbf{x}_t | \mathbf{w}_t, \mathbf{M}) \rangle_{q(\mathbf{w}_t)\, q(\mathbf{M})} - \mathrm{D}_{\mathrm{KL}}(q(\mathbf{w}_t) \| p(\mathbf{w}_t)) \right) - \mathrm{D}_{\mathrm{KL}}(q(\mathbf{M}) \| p(\mathbf{M})) \quad (4)$$

For consistency, we write $p(\mathbf{w}_t) = p(\mathbf{w}) = \mathcal{N}(\mathbf{0}, \mathbf{1})$. From the perspective of the EM algorithm [11], the lower-bound can be maximised in two ways: 1. By tightening the the bound while keeping the likelihood unchanged. This can be achieved by minimising the KL-divergences in eq. 3, so that $q(\mathbf{w}_t)$ approximates the posterior distribution $p(\mathbf{w}_t | \mathbf{x}_t, \mathbf{M})$ and $q(\mathbf{M})$ approximates the posterior distribution $p(\mathbf{M} | \mathbf{x}_{\leqslant T})$. 2. By directly maximising the lower-bound $\mathcal{L}_T$ as an evidence lower-bound objective (ELBO) by, for example, gradient ascent on parameters of $\mathcal{L}_T$[1]. This may both improve the quality of posterior approximation by squeezing the bound, and maximising the likelihood of the generative model.

We develop an algorithm analogous to the two step-EM algorithm: it first analytically tighten the lower-bound by minimising the KL-divergence terms in eq. 3 via inference of tractable parameters, and then maximises the lower-bound by slow updating of the remaining model parameters via backpropagation. The analytic inference in the first step is quick and does not require training, allowing the model to adapt to new data at test time.

### 3.1 Dynamic Addressing

Recall that the approximate posterior distribution of $\mathbf{w}_t$ has the form: $q(\mathbf{w}_t) = \mathcal{N}\left(\mu_{\mathbf{w}_t}, \sigma_{\mathbf{w}}^2\right)$. While the variance parameter is trained using gradient-based updates, dynamic addressing is used to find the $\mu_{w_t}$ that minimises $\mathrm{D}_{\mathrm{KL}}(q(\mathbf{w}) \| p(\mathbf{w} | \mathbf{x}, \mathbf{M}))$. Dropping the subscript when it applies to any given $\mathbf{x}$ and $\mathbf{M}$, it can be shown that the KL-divergence can be approximated by the following quadratic form (see Appendix D for derivation):

$$\mathrm{D}_{\mathrm{KL}}(q(\mathbf{w}) \| p(\mathbf{w} | \mathbf{x}, \mathbf{M})) \approx -\frac{\|e(\mathbf{x}) - \mathbf{M}^{\mathsf{T}} \mu_{\mathbf{w}}\|^2}{2\sigma_{\xi}^2} - \frac{1}{2} \|\mu_{\mathbf{w}}\|^2 + \ldots \quad (5)$$

where the terms that are independent of $\mu_w$ are omitted. Then, the optimal $\mu_{\mathbf{w}}$ can be found by solving the (regularised) least-squares problem:

$$\mu_w \leftarrow \left(\mathbf{M}\,\mathbf{M}^{\mathsf{T}} + \sigma_\xi^2 \cdot \mathbf{I}\right)^{-1} \mathbf{M}^{\mathsf{T}}\, e(\mathbf{x}) \tag{6}$$

This operation can be implemented efficiently via an off-the-shelve least-square solver, such as TensorFlow's `matrix_solve_ls` function which we used in experiments. Intuitively, dynamic addressing finds the combination of memory rows that minimises the square error between the read out $\mathbf{M}^{\mathsf{T}}\mu_{\mathbf{w}}$ and the embedding $\mathbf{z} = e(\mathbf{x})$, subject to the constraint from the prior $p(\mathbf{w})$.

### 3.2  Bayesian Memory Update

We now turn to the more challenging problem of minimising $\mathrm{D}_{\mathrm{KL}}\left(q\left(\mathbf{M}\right)\|p\left(\mathbf{M}|\mathbf{x}_{\leqslant T}\right)\right)$. We tackle this minimisation via a sequential update algorithm. To motivate this algorithm we begin by considering $T = 1$. In this case, eq. 3 can be simplified to:

$$\ln p\left(\mathbf{x}_1\right) = \mathcal{L}_1 + \left\langle \mathrm{D}_{\mathrm{KL}}\left(q\left(\mathbf{w}_1\right)\|p\left(\mathbf{w}_1|\mathbf{x}_1,\mathbf{M}\right)\right)\right\rangle_{q(\mathbf{M})} + \mathrm{D}_{\mathrm{KL}}\left(q\left(\mathbf{M}_1\right)\|p\left(\mathbf{M}_1|\mathbf{x}_1\right)\right) \tag{7}$$

While it is still unclear how to minimise $\mathrm{D}_{\mathrm{KL}}\left(q\left(\mathbf{M}_1\right)\|p\left(\mathbf{M}|\mathbf{x}_1\right)\right)$, *if a suitable weight distribution $q\left(\mathbf{w}_1\right)$ were given*, a slightly different term $\mathrm{D}_{\mathrm{KL}}\left(q\left(\mathbf{M}_1|\mathbf{w}_1\right)\|p\left(\mathbf{M}_1|\mathbf{w}_1,\mathbf{x}_1\right)\right)$ can be minimised to 0. To achieve this, we can set $q\left(\mathbf{M}_1|\mathbf{w}_1\right) = p\left(\mathbf{M}_1|\mathbf{x}_1,\mathbf{w}_1\right)$ by updating the parameters of $q(\mathbf{M})$ using the same Bayesian update rule as in the KM (Appendix A): $\mathbf{R}_1, \mathbf{U}_1 \leftarrow \mathbf{R}_0, \mathbf{U}_0$. We may then marginalise out $\mathbf{w}_1$ to obtain

$$q\left(\mathbf{M}_1\right) = \int q\left(\mathbf{M}_1|\mathbf{w}_1\right) q\left(\mathbf{w}_1\right)\,\mathrm{d}\mathbf{w}_1 \tag{8}$$

A reasonable *guess* of $\mathbf{w}_1$ can be obtained by be solving

$$q\left(\mathbf{w}_1\right) \leftarrow \underset{q'(\mathbf{w}_1)}{\operatorname{argmax}} \mathrm{D}_{\mathrm{KL}}\left(q'(\mathbf{w}_1)\|p\left(\mathbf{w}_1|\mathbf{x}_1,\mathbf{M}_0\right)\right) \tag{9}$$

as in section 3.1, but using the prior memory $\mathbf{M}_0$. To continue, we treat the current posterior $q\left(\mathbf{M}_1\right)$ as next prior, and compute $q\left(\mathbf{M}_2\right)$ using $\mathbf{x}_2$ following the same procedure until we obtain $q\left(\mathbf{M}_T\right)$ using all $\mathbf{x}_{\leqslant T}$.

More formally, Appendix C shows this heuristic online update procedure maximises another lower-bound of the log-likelihood. In addition, the marginalisation in eq. 8 can be approximated by using $\mu_{\mathbf{w}_t}$ instead of sampling $\mathbf{w}_t$ for each memory update:

$$q\left(\mathbf{M}_t\right) \approx p\left(\mathbf{M}_t|\mathbf{x}_t, \mu_{\mathbf{w}_t}\right) \tag{10}$$

Although this lower-bound is looser than $\mathcal{L}_T$ (eq. 4), Appendix C suggests it can be tighten by iteratively using the updated memory for addressing (e.g., replacing $M_0$ in eq. 9 by the updated $M_1$, the "optional" step in Algorithm 1) and update the memory with the refined $q\left(\mathbf{w}_t\right)$. We found that extra iterations yielded only marginal improvement in our setting, so we did not use it in our experiments.

### 3.3  Gradient-Based Training

Having inferred $q\left(\mathbf{w}_t\right)$ and $q\left(\mathbf{M}\right) = q\left(\mathbf{M}_T\right)$, we now focus on gradient-based optimisation of the lower-bound $\mathcal{L}_T$ (eq. 4). To ensure the likelihood in eq. 4 $\ln p\left(\mathbf{x}|\mathbf{w},\mathbf{M}\right)$ can be produced from the likelihood given by the memory $\ln p\left(\mathbf{z}|\mathbf{w},\mathbf{M}\right)$, we ideally need a bijective pair of encoder and decoder $\mathbf{x} \Longleftrightarrow \mathbf{z}$ (see Appendix D for more discussion). This is difficult to guarantee, but we can approximate this condition by maximising the autoencoder log-likelihood:

$$\mathcal{L}_{\mathrm{AE}} = \left\langle \ln d\left(e(\mathbf{x})\right)\right\rangle_{\mathbf{x}\sim\mathcal{D}} \tag{11}$$

Taken together, we maximise the following joint objective using backpropagation:

$$\mathcal{O} = \mathcal{L}_T + \mathcal{L}_{\mathrm{AE}} \tag{12}$$

We note that dynamic addressing during online memory updates introduces order dependence since $q\left(\mathbf{w}_t\right)$ always depends on the previous memory. This violates the model's exchangeable structure (order-independence). Nevertheless, gradient-ascend on $\mathcal{L}_T$ mitigates this effect by adjusting the model so that $\mathrm{D}_{\mathrm{KL}}\left(q\left(\mathbf{w}\right)\|p\left(\mathbf{w}|\mathbf{x},\mathbf{M}\right)\right)$ remains close to a minimum even for previous $q\left(\mathbf{w}_t\right)$. Appendix C explains this in more details.

## 3.4 Prediction / Reading

The predictive distribution of our model is the posterior distribution of $\mathbf{x}$ given a query $\mathbf{x}_q$ and memory $\mathbf{M}$: $p\left(\mathbf{x}|\mathbf{x}_q, \mathbf{M}\right) = \int p\left(\mathbf{x}|\mathbf{w}, \mathbf{M}\right) p\left(\mathbf{w}|\mathbf{x}_q, \mathbf{M}\right) d\mathbf{w}$ This posterior distribution does not have an analytic form in general (unless $d(\mathbf{x}|\mathbf{z})$ is Gaussian). We therefore approximate the integral using the *maximum a posteriori* (MAP) estimator of $\mathbf{w}^*$:

$$p\left(\hat{\mathbf{x}}|\mathbf{x}_q, \mathbf{M}\right) = p\left(\mathbf{x}|\mathbf{w}^*, \mathbf{M}\right)\Big|_{\mathbf{w}^* = \mathrm{argmax}_{\mathbf{w}} \, p(\mathbf{w}|\mathbf{x}_q, \mathbf{M})} \tag{13}$$

Thus, $\mathbf{w}^*$ can be computed by solving the same least-square problem as in eq. 6 and choosing $\mathbf{w}^* = \mu_{\mathbf{w}}$ (see Appendix D for details).

## 3.5 Attractor Dynamics

To understand the model's attractor dynamics, we define the *energy* of a configuration $(\mathbf{x}, \mathbf{w})$ with a given memory $\mathbf{M}$ as:

$$\mathcal{E}(\mathbf{x}, \mathbf{w}) = -\left\langle \ln p\left(\mathbf{x}|\mathbf{w}, \mathbf{M}\right)\right\rangle_{q(\mathbf{M})} + \mathrm{D_{KL}}\left(q_t(\mathbf{w})\|p\left(\mathbf{w}\right)\right) \tag{14}$$

For a well trained model, with $\mathbf{x}$ fixed, $\mathcal{E}(\mathbf{x}, \mathbf{w})$ is at minimum with respect to $\mathbf{w}$ after minimising $\mathrm{D_{KL}}\left(q\left(\mathbf{w}\right)\|p\left(\mathbf{w}|\mathbf{x}, \mathbf{M}\right)\right)$ (eq. 6). To see this, note that the negative of $\mathcal{E}(\mathbf{x}, \mathbf{w})$ consist of just terms in $\mathcal{L}_T$ in eq. 4 that depend on a specific $\mathbf{x}$ and $\mathbf{w}$, which are maximised during training. Now we can minimise $\mathcal{E}(\mathbf{x}, \mathbf{w})$ further by fixing $\mathbf{w}$ and optimising $\mathbf{x}$. Since only the first term depends on $\mathbf{x}$, $\mathcal{E}(\mathbf{x}, \mathbf{w})$ is further minimised by choosing the mode of the likelihood function $\left\langle \ln p\left(\mathbf{x}|\mathbf{w}, \mathbf{M}\right)\right\rangle_{q(\mathbf{M})}$. For example, we take the mean for the Gaussian likelihood, and round the sigmoid outputs for the Bernoulli likelihood. Each step can be viewed as coordinate descent over the energy $\mathcal{E}(\mathbf{x}, \mathbf{w})$, as illustrated in figure 2 (left).

The step of optimising $\mathbf{w}$ following by taking the mode of $\mathbf{x}$ is exactly the same as taking the mode of the predictive distribution $\mathbf{x}_{\mathrm{mode}} = \mathrm{argmax}_{\hat{\mathbf{x}}} \, p\left(\hat{\mathbf{x}}|\mathbf{x}_q, \mathbf{M}\right)$ (eq. 13). Therefore, we can simulate the attractor dynamics by repeatedly feeding-back the predictive mode as the next query: $\mathbf{x}_1 = \mathrm{argmax}_{\hat{\mathbf{x}}} \, p(\hat{\mathbf{x}}|\mathbf{x}_0, \mathbf{M}), \mathbf{x}_2 = \mathrm{argmax}_{\hat{\mathbf{x}}} \, p(\hat{\mathbf{x}}|\mathbf{x}_1, \mathbf{M}), \ldots, \mathbf{x}_n = \mathrm{argmax}_{\hat{\mathbf{x}}} \, p(\hat{\mathbf{x}}|\mathbf{x}_{n-1}, \mathbf{M})$. This sequence converges to a stored pattern in the memory, because each iteration minimises the energy $\mathcal{E}(\mathbf{x}, \mathbf{w})$, so that $\mathcal{E}(\mathbf{x}_n, \mathbf{w}_n) < \mathcal{E}(\mathbf{x}_{n-1}, \mathbf{w}_{n-1})$, unless it has already converged at $\mathcal{E}(\mathbf{x}_n, \mathbf{w}_n) = \mathcal{E}(\mathbf{x}_{n-1}, \mathbf{w}_{n-1})$. Therefore, the sequence will converge to some $\mathbf{x}^*$, a local minimum in the energy landscape, which in a well trained memory model corresponds to a stored pattern.

Viewing $\mathbf{x}_n = \mathrm{argmax}_{\hat{\mathbf{x}}} \, p(\hat{\mathbf{x}}|\mathbf{x}_{n-1}, \mathbf{M})$ as a dynamical system, the stored patterns correspond to point attractors in this system. See Appendix C for a formal treatment. In this work we employed deterministic dynamics in our experiments and to simplify analysis. Alternatively, sampling from $q\left(\mathbf{w}\right)$ and the predictive distribution would give stochastic dynamics that simulate Markov-Chain Monte Carlo (MCMC). We leave this direction for future investigation.

# 4 Experiments

We tested our model on Ominglot [22] and frames from DMLab tasks [6]. Both datasets have images from a large number of classes, well suited to testing fast adapting external memory: 1200 different characters in Omniglot, and infinitely many procedurally generated 3-D maze environments from DMLab. We treat Omniglot as binary data, while DMLab has larger real-valued colour images. We demonstrate that the same model structure with identical hyperparameters (except for number of filters, the predictive distribution, and memory size) can readily handle these different types of data.

To compare with the KM, we followed [32] to prepare the Ominglot dataset, and employed the same convolutional encoder and decoder structure. We trained all models using the Adam optimiser [19] with learning rate $1 \times 10^{-4}$. We used 16 filters in the convnet and $32 \times 100$ memory for Omniglot, and 256 filters and $64 \times 200$ memory for DMLab. We used the Bernoulli likelihood function for Omniglot, and the Gaussian likelihood function for DMLab data. Uniform noise $\mathcal{U}(0, \frac{1}{128})$ was added to the labyrinth data to prevent the Gaussian likelihood from collapsing.

Following [32], we report the lower-bound on the conditional log-likelihood $\ln p\left(\mathbf{x}_{\leqslant T}|\mathbf{M}\right)$ by removing $\mathrm{D_{KL}}\left(q\left(\mathbf{M}\right)\|p\left(\mathbf{M}\right)\right)$ from $\mathcal{L}_T$ (eq. 4). This is the negative energy $-\mathcal{E}$, and we obtained

the per-image bound (i.e., conditional ELBO) by dividing it by the episode size. We trained the model for Omniglot for approximately $3 \times 10^5$ steps; the test conditional ELBO reached 77.2, which is worse than the 68.3 reported from the KM [32]. However, we show that the DKM generalises much better to unseen long episodes. We trained the model for DMLab for $1.1 \times 10^5$ steps; the test conditional ELBO reached $-9046.5$, which corresponds to 2.75 bits per pixel. After training, we used the same testing protocol as [32], first computing the posterior distribution of memory (writing) given an episode, and then performing tasks using the memory's posterior mean. For reference, our implementation of the memory module is provided at `https://github.com/deepmind/dynamic-kanerva-machines`.

**Capacity**

We investigated memory capacity using the Omniglot dataset, and compared our model with the KM and DNC. To account for the additional $\mathcal{O}(K^3)$ cost in the proposed dynamic addressing, our model in Omniglot experiments used a significantly smaller number of memory parameters ($32 \times 100 + 32 \times 32$) than the DNC ($64 \times 100$), and less than half of that used for the KM in [32]. Moreover, our model does not have additional parametrised structure, like the memory controllers in DNC or the amortised addressing module in the KM. As in [32], we train our model using episodes with 32 patterns randomly sampled from all classes, and test it using episodes with lengths ranging from 10 to 200, drawn from 2, 4, or 8 classes of characters (i.e. varying the redundancy of the observed data). We report retrieval error as the negative of the conditional ELBO. The results are shown in figure 2 (right), with results for the KM and DNC adapted from [32].

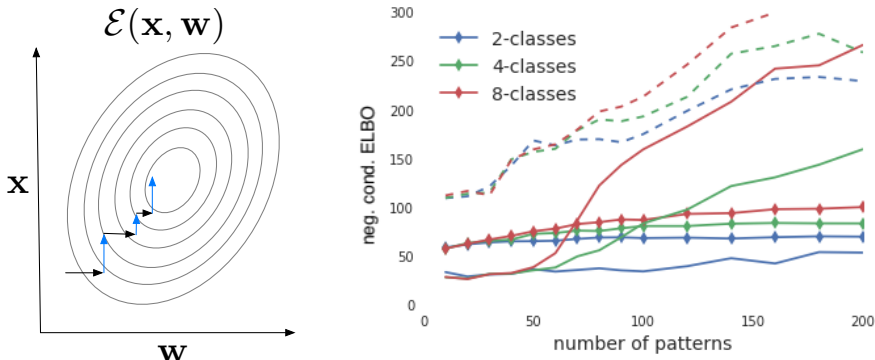

Figure 2: Left: Illustration of the attractor dynamics that converge to a local minimum of the energy $\mathcal{E}(\mathbf{x}, \mathbf{w})$. The circles shows contours of the energy. Black arrows shows the results from optimising $\mathbf{w}$ by solving the least-square problem; blue arrows depict optimisation of $\mathbf{x}$ by taking the mode of the predictive distribution. Right: Comparing the capacity of our model (diamond lines) with the KM (solid lines) and the DNC (dashed lines). Our model compresses and generalises significantly better for long episodes.

The capacity curves for our model are strikingly flat compared with both the DNC and the KM; we believe that this is because the parameter-free addressing (section 3.1) generalises to longer episodes much better than the parametrised addressing modules in the DNC or the KM. The errors are larger than the KM for small numbers of patterns (approximately <60), possibly because the KM over-fits to shorter episodes that were more similar to training episodes.

**Attractor Dynamics: Denoising and Sampling**

We next verified the attractor dynamics through denoising and sampling tasks. These task demonstrate how low-quality patterns, either from noise-corruption or imperfect priors, can be corrected using the attractor dynamics. Figure 3 (**a**) and Figure 4 (**a**) show the result of denoising. We added salt-and-pepper noise to Omniglot images by randomly flipping 15% of the bits, and independent Gaussian noise $\mathcal{N}(0, 0.15)$ to all pixels in DMLab images. Such noise is never presented during training. We ran the attractor dynamics (section 3.5) for 15 iterations from the noise corrupted images. Despite the significant corruption of images via different types of noise, the image quality improved steadily for both datasets. Interestingly, the denoised Omniglot patterns are even cleaner and smoother than the

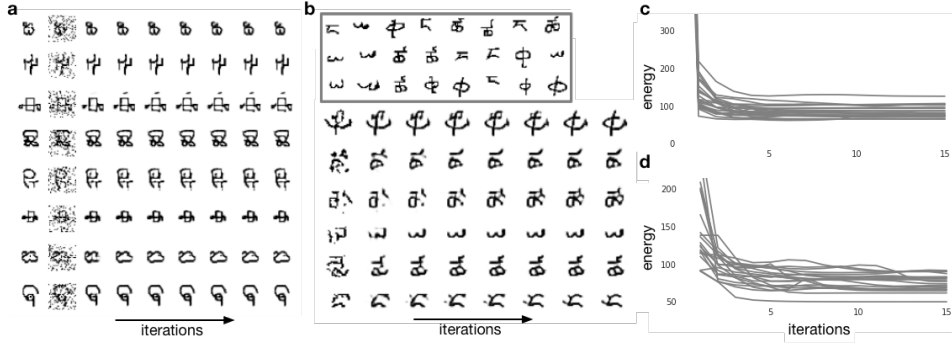

Figure 3: **a**: Denoising of Omniglot patterns. Patterns in the second column are obtained by adding $15\%$ salt-and-pepper noise to the patterns in the first column. The following columns shows samples from consecutive iterations. **b**: Sampling of Omniglot patterns. Patterns inside the top box were written into memory. Patterns in the first column are reconstructed using $w \sim p(w)$, which are then improved through iterations in the following columns. **c**: Energy as a function of iterations during denoising. **d**: Energy as a function of iterations during sampling.

original patterns. The trajectories of the energy during denoising for 20 examples (including those we plotted as images) are shown in Figure 3 (**c**) and Figure 4 (**c**), demonstrating that the system states were attracted to points with lower energy.

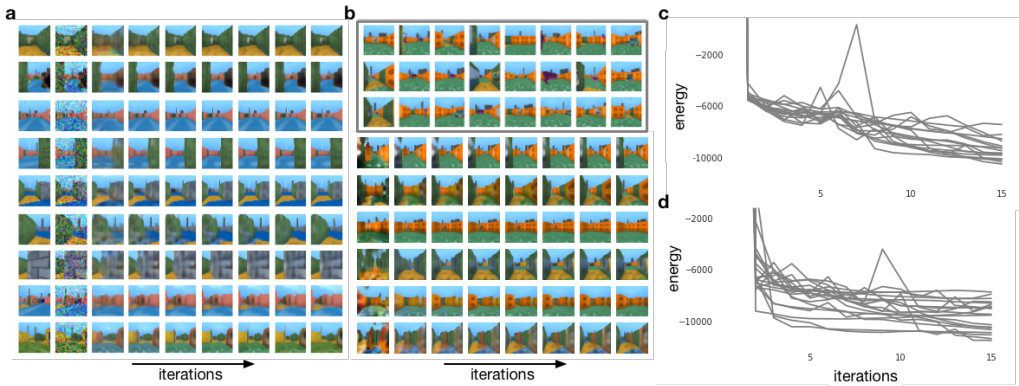

Figure 4: Experiment results for DMLab. Description of each panel is matched to those in Figure 3.

Sampling from the models' prior distributions provides another application of the attractor dynamics. Generative models trained with stochastic variational inference usually suffer from the problem of low sample quality, because the asymmetric KL-divergence they minimise usually results in priors broader than the posterior that is used to train the decoders. While different approaches exist to improve sample quality, including using more elaborated posteriors [26] and different training objectives [13], our model solves this problem by moving to regions with higher likelihoods via the attractor dynamics. As illustrated in Figure 3 (**c**) and Figure 4 (**c**), the initial samples have relatively low quality, but they were improved steadily through iterations. This improvement is correlated with the decrease of energy. We do observe fluctuations in energy in all experiments, especially for DMLab. This may be caused by the saddle-points that are more common in larger models [9]. While the observation of saddle-points violates our assumption of local minima (section 3.5), our model still worked well and the energy generally dropped after temporarily rising.

# 5 Discussion

Here we have presented a novel approach to robust attractor dynamics inside a generative distributed memory. Other than the neural network encoder and decoder, our model has only a small number of statistically well-defined parameters. Despite its simplicity, we have demonstrated its high capacity by efficiently compressing episodes online and have shown its robustness in retrieving patterns corrupted by unseen noise.

Our model can trade increased computation for higher precision retrieval by running attractor dynamics for more iterations. This idea of using attractors for memory retrieval and cleanup dates to Hopfield nets [15] and Kanerva's sparse distributed memory [18]. Zemel and Mozer proposed a generative model for memory [33] that pioneered the use of variational free energy to construct attractors for memory. By restricting themselves to a localist representation, their model is easy to train without backpropagation, though this choice constrains its capacity. On the other hand, Boltzmann Machines [1] are high-capacity generative models with distributed representations which obey stochastic attractor dynamics. However, writing memories into the weights of Boltzmann machines is typically slow and difficult. In comparison, the DKM trains quickly via a low-variance gradient estimator and allows fast memory writing as inference. Notably, Saul and Jordan [30] discussed the limit of undirected graphical models compared with directed models in learning, and proposed mean-field-based attractor dynamics for iterative inference in feed-forward belief networks. Our method can also be seen as an extension along this line.

As a principled probabilistic model, the linear Gaussian memory of the DKM can be seen as a special case of the Kalman Filter (KF) [17] without the drift-diffusion dynamics of the latent state. This more stable structure captures the statistics of entire episodes during sequential updates with minimal interference. The idea of using the latent state of the KF as memory is closely related to the hetero-associative novelty filter suggested in [10]. The DKM can be also contrasted with recently proposed nonlinear generalisations of the KF such as [21] in that we preserve the higher-level linearity for efficient analytic inference over a very large latent state ($\mathbf{M}$). By combining deep neural networks and variational inference this allows our model to store associations between a large number of patterns, and generalise to large scale non-Gaussian datasets .

## Footnotes

[1]This differs from the original EM algorithm, which fixes the approximated posterior in the M step.

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
