[Supplementary Material]

# Learning Attractor Dynamics
# for Generative Memory (Appendix)

**Yan Wu, Greg Wayne, Karol Gregor, Timothy Lillicrap**
DeepMind
{yanwu,gregwayne,karolg,countzero}@google.com

## A   The Bayesian update rule for the Kanerva Machine

Here we reproduce the exact Bayesian update rule used in the Kanerva Machine.

$$p\left(\mathbf{M}|\mathbf{z},\mathbf{w}\right) = \frac{p\left(\mathbf{z}|\mathbf{w},\mathbf{M}\right)\,p\left(\mathbf{M}\right)}{\int p\left(\mathbf{z}|\mathbf{w},\mathbf{M}\right)\,p\left(\mathbf{M}\right)\,\mathrm{d}\mathbf{M}} \tag{1}$$

A memory with mean $R_{t-1}$, row covariance $U_{t-1}$ and observational noise variance $\sigma_\xi^2$ is updated given a newly observed sample $z$ and its addressing weight $w$ by:

$$\boldsymbol{\Delta} \leftarrow \mathbf{z} - \mathbf{R}_{t-1}^{\intercal}\mathbf{w} \tag{2}$$

$$\boldsymbol{\Sigma}_c \leftarrow \mathbf{U}_{t-1}\,\mathbf{w} \tag{3}$$

$$\boldsymbol{\Sigma}_{\mathbf{z}} \leftarrow \mathbf{w}^{\intercal}\,\mathbf{U}_{t-1}\,\mathbf{w} + \sigma_\xi^2 \tag{4}$$

$$\mathbf{R}_t \leftarrow \mathbf{R}_{t-1} + \boldsymbol{\Sigma}_c\,\boldsymbol{\Sigma}_{\mathbf{z}}^{-1}\,\boldsymbol{\Delta}^{\intercal} \tag{5}$$

$$\mathbf{U}_t \leftarrow \mathbf{U}_{t-1} - \boldsymbol{\Sigma}_c\,\boldsymbol{\Sigma}_{\mathbf{z}}^{-1}\,\boldsymbol{\Sigma}_c^{\intercal} \tag{6}$$

Note that the new covariance $U_t$ would collapse to zero if the noise variance $\sigma^2 \to 0$.

The graphical model assumes an observational noise $\mathcal{N}\left(\mathbf{0},\sigma_\xi^{\mathbf{2}}\right)$, which results in the read out distribution given the addressing weights $\mathbf{w}$, $p\left(\mathbf{z}|\mathbf{R}\,\mathbf{w},\sigma_\xi^2\right)$ (eq. 1 in matrix notation). We ignore observation noise when reading the memory and directly take $\mathbf{z} \leftarrow \mathbf{R}\,\mathbf{w}$. This simplification reduces the variance in training and is justified by the fact that the fixed observation noise does not convey any information. A strong-enough decoder will learn to remove such noise through training.

## B   Parameters and Initialisation

Here we enumerate parameters of the model and their initialisations. In our experiments, the memory parameters are insensitive to initial values.

| Parameters | description | Initial Value |
|---|---|---|
| $\mathbf{R}_0$ | $K \times C$ memory prior mean matrix | $\mathcal{N}\left(\mathbf{0},\mathbf{1}\right)$ |
| $\mathbf{U}_0 = \sigma_U^2\,\mathbf{I}$ | $K \times K$ memory prior covariance matrix | $\sigma_U^2 = 1.0$ |
| $\sigma_{\mathbf{w}}^2$ | Addressing weight posterior variance | 0.3 |
| $\sigma_\xi^2$ | Memory observation noise variance | 1.0 |
| $\ldots$ | Neural network weights of encoder and decoder | Glorot Initialization |

Figure 1: The probabilistic graphical model illustrating sequential online updates of memory. Dashed lines show the inference model and solid lines illustrate the generative model. For brevity, we illustrated only the inference of $\mathbf{M}_1$ given $\mathbf{M}_0$, $\mathbf{w}_1$ and $\mathbf{x}_1$; all the following steps are the same until reaching the end of the episode $\mathbf{x}_T$.

## C  Sequential Variational Inference for Memory

The log-likelihood for any $\mathbf{x}_{\leqslant T}$ can be decompose as a sum of a variational lower-bound and KL-divergences as:

$$
\begin{aligned}
\ln p\left(\mathbf{x}_{\leqslant T}\right) &= \ln \frac{p\left(\mathbf{x}_{\leqslant T}, \mathbf{w}_{\leqslant T}, \mathbf{M}\right)}{p\left(\mathbf{w}_{\leqslant T}, \mathbf{M}|\mathbf{x}_{\leqslant T}\right)} \\
&= \left\langle \ln \frac{p\left(\mathbf{x}_{\leqslant T}|\mathbf{w}_{\leqslant T}, \mathbf{M}\right) \, p\left(\mathbf{w}_{\leqslant T}\right) \, p\left(\mathbf{M}\right) \, q\left(\mathbf{w}_{\leqslant T}\right) \, q\left(\mathbf{M}\right)}{p\left(\mathbf{w}_{\leqslant T}|\mathbf{x}_{\leqslant T}, \mathbf{M}\right) \, p\left(\mathbf{M}|\mathbf{x}_{\leqslant T}\right) \, q\left(\mathbf{w}_{\leqslant T}\right) \, q\left(\mathbf{M}\right)} \right\rangle_{q(\mathbf{M})\, q\left(\mathbf{w}_{\leqslant T}\right)} \\
&= \mathcal{L}_T + \sum_{t=1}^{T} \left\langle \mathrm{D}_{\mathrm{KL}}\left(q\left(\mathbf{w}_t\right) \| p\left(\mathbf{w}_t|\mathbf{x}_t, \mathbf{M}\right)\right)\right\rangle_{q(\mathbf{M})} + \mathrm{D}_{\mathrm{KL}}\left(q\left(\mathbf{M}\right) \| p\left(\mathbf{M}|\mathbf{x}_{\leqslant T}\right)\right)
\end{aligned} \tag{7}
$$

$$
\mathcal{L}_T = \sum_{t=1}^{T} \left( \left\langle \ln p\left(\mathbf{x}_t|\mathbf{w}_t, \mathbf{M}\right)\right\rangle_{q(\mathbf{w}_t)\, q(\mathbf{M})} - \mathrm{D}_{\mathrm{KL}}\left(q\left(\mathbf{w}_t\right) \| p\left(\mathbf{w}_t\right)\right)\right) - \mathrm{D}_{\mathrm{KL}}\left(q\left(\mathbf{M}\right) \| p\left(\mathbf{M}\right)\right) \tag{8}
$$

However, as we noted in the main text, it is hard to maximise $\mathcal{L}_T$ directly, since we can not compute $q\left(\mathbf{M}\right) \approx p\left(\mathbf{M}|\mathbf{x}_{\leqslant T}\right)$ directly.

To derive a sequential update rule of the memory to compute $q\left(\mathbf{M}\right) \approx p\left(\mathbf{M}|\mathbf{x}_{\leqslant T}\right)$, we consider updating the memory for step $t$ of an episode. This assumes memory from the previous update $q\left(\mathbf{M}_{t-1}; \mathbf{R}_{t-1}, \mathbf{U}_{t-1}\right) \approx p\left(\mathbf{M}_{t-1}|\mathbf{x}_{\leqslant t-1}\right)$ is given, so that we can decompose $\ln p\left(\mathbf{x}_{\leqslant t}\right)$ conditioned on $\mathbf{M}_{t-1}$:

$$
\begin{aligned}
\ln p\left(\mathbf{x}_{\leqslant t}\right) &= \underbrace{\left\langle \ln p\left(\mathbf{x}_{\leqslant t}|\mathbf{M}_{t-1}\right)\right\rangle_{q(\mathbf{M}_{t-1})} - \mathrm{D}_{\mathrm{KL}}\left(q\left(\mathbf{M}_{t-1}\right) \| p\left(\mathbf{M}_{t-1}\right)\right)}_{\mathcal{L}_t} \\
&\quad + \mathrm{D}_{\mathrm{KL}}\left(q\left(\mathbf{M}_{t-1}\right) \| p\left(\mathbf{M}_{t-1}|\mathbf{x}_{\leqslant t}\right)\right)
\end{aligned} \tag{9}
$$

where we have a likelihood lower-bound $\mathcal{L}_t$, which has the same form as $\mathcal{L}_T$. This lower-bound is tight when $q\left(\mathbf{M}_{t-1}\right) = p\left(\mathbf{M}_{t-1}|\mathbf{x}_{\leqslant t}\right)$. This suggests, ideally, that the memory $q\left(\mathbf{M}_{t-1}\right)$ at step $t-1$ needs to be *predictive* of the next observation $\mathbf{x}_t$, in addition to accumulating information from the $\mathbf{x}_{\leqslant t-1}$ that are already used in computing $q\left(\mathbf{M}_{t-1}\right)$.

As illustrated in Figure 1, we assume a deterministic transition $q\left(\mathbf{M}_t|\mathbf{M}_{t-1}\right) = \delta(\mathbf{M}_{t-1})$, so the prior of $q\left(\mathbf{M}_t\right)$ simplifies to $\int q\left(\mathbf{M}_t|\mathbf{M}_{t-1}\right) q\left(\mathbf{M}_{t-1}\right) \mathrm{d}\mathbf{M}_{t-1} = \int \delta(\mathbf{M}_{t-1}) q\left(\mathbf{M}_{t-1}\right) \mathrm{d}\mathbf{M}_{t-1} = q\left(\mathbf{M}_{t-1}\right)$. We can then *recursively* expand the likelihood term in eq. 9, similar to eq. 7 and eq. 8 (omitting the expectation over $q\left(\mathbf{M}_{t-1}\right)$):

$$
\begin{aligned}
\ln p\left(\mathbf{x}_{\leqslant t}|\mathbf{M}_{t-1}\right) &= \mathcal{L}'_t + \sum_{t'=1}^{t} \mathrm{D}_{\mathrm{KL}}\left(q\left(\mathbf{w}_{t'}\right) \| p\left(\mathbf{w}_{t'}|\mathbf{x}_{t'}, \mathbf{M}_{t-1}\right)\right) \\
&\quad + \mathrm{D}_{\mathrm{KL}}\left(q\left(\mathbf{M}_t|\mathbf{w}_t\right) \| p\left(\mathbf{M}_t|\mathbf{x}_{\leqslant t}, \mathbf{w}_{\leqslant t}\right)\right)
\end{aligned} \tag{10}
$$

$$\mathcal{L}'_t = \sum_{t'=1}^{t} \left( \langle \ln p\left(\mathbf{x}_{t'}|\mathbf{w}_{t'}, \mathbf{M}_t\right)\rangle_{q(\mathbf{w}_{t'}), q(\mathbf{M}_t|\mathbf{w}_t)} - \mathrm{D_{KL}}\left(q\left(\mathbf{w}_{t'}\right) \| p\left(\mathbf{w}_{t'}\right)\right)\right)$$
$$- \langle \mathrm{D_{KL}}\left(q\left(\mathbf{M}_t|\mathbf{w}_t\right) \| q\left(\mathbf{M}_{t-1}\right)\right)\rangle_{q(\mathbf{w}_t)} \tag{11}$$

The above $\mathcal{L}'_t$ is easier to maximise, since it only depends on $\mathbf{x}_t$, and on $\mathbf{M}_{t-1}$, which we assume we know. We can minimise the gap between $\mathcal{L}'_t$ and $\ln p\left(\mathbf{x}_{\leqslant t}|\mathbf{M}_{t-1}\right)$ by minimising $\mathrm{D_{KL}}\left(q\left(\mathbf{M}_t|\mathbf{w}_t\right) \| p\left(\mathbf{M}_t|\mathbf{x}_{\leqslant t}, \mathbf{w}_{\leqslant t}\right)\right)$ using the Bayes' update rule (Appendix A), and minimising $\sum_{t'=1}^{t}\mathrm{D_{KL}}\left(q\left(\mathbf{w}_{t'}\right) \| p\left(\mathbf{w}_{t'}|\mathbf{x}_{t'}, \mathbf{M}_{t-1}\right)\right)$ using dynamic addressing (Section 3.1).

We can tighten $\mathcal{L}_t$ by allowing further updating iterations as shown by the optional step in Algorithm 1. This is likely to be a tighter lower-bound, since generally the KL-divergence $\mathrm{D_{KL}}\left(q\left(\mathbf{M}_t\right) \| p\left(\mathbf{M}_{t-1}|\mathbf{x}_{\leqslant t}\right)\right) < \mathrm{D_{KL}}\left(q\left(\mathbf{M}_{t-1}\right) \| p\left(\mathbf{M}_{t-1}|\mathbf{x}_{\leqslant t-1}\right)\right)$ after incorporating information from $\mathbf{x}_t$. This process can be repeated until this KL-divergence is tightened to it's minimum.

From the above equations, we have the inequality

$$\ln p\left(\mathbf{x}_{\leqslant t}\right) \geqslant \mathcal{L}_t \geqslant \underbrace{\mathcal{L}'_t - \mathrm{D_{KL}}\left(q\left(\mathbf{M}_{t-1}\right) \| p\left(\mathbf{M}_{t-1}\right)\right)}_{\mathcal{B}_t} \tag{12}$$

Therefore, we can maximising $\ln p\left(\mathbf{x}_{\leqslant t}\right)$ by maximising the lower-bound $\mathcal{B}_t$. Naively, in eq. 10, all the $t$ terms in $\sum_{t'=1}^{t}\mathrm{D_{KL}}\left(q\left(\mathbf{w}_{t'}\right) \| p\left(\mathbf{w}_{t'}|\mathbf{x}_{t'}, \mathbf{M}_{t-1}\right)\right)$ need to be minimised at step $t$. This would result in $\mathcal{O}(T^2)$ cost in both inferring $q\left(\mathbf{w}_{\leqslant T}\right)$ and $q\left(\mathbf{M}_T\right)$. To reduce the computational cost, we keep previously $q\left(\mathbf{w}_{\leqslant t-1}\right)$, and only infer $q\left(\mathbf{w}_t\right)$, resulting in only $\mathcal{O}(T)$ cost. The trade-off is a looser lower-bound $\mathcal{L}'_t$ and therefore a looser $\mathcal{B}_t$, since some of the $\mathrm{D_{KL}}\left(q\left(\mathbf{w}_{t'}\right) \| p\left(\mathbf{w}_{t'}|\mathbf{x}_{t'}, \mathbf{M}_{t-1}\right)\right)$ for $t' < t$ may not be minimised.

Once $q\left(\mathbf{M}_t|\mathbf{w}_t\right)$ is computed, we can compute the marginal for the next step as:

$$q\left(\mathbf{M}_t\right) = \int q\left(\mathbf{M}_t|\mathbf{w}_t\right) q\left(\mathbf{w}_t\right) \mathrm{d}\mathbf{w}_t$$
$$\approx q\left(\mathbf{M}_t|\mathbf{w}_t\right)\Big|_{\mathbf{w}^*=\mathrm{argmax}_\mathbf{w}\, p(\mathbf{w}|\mathbf{x}_q, \mathbf{M})} \tag{13}$$

Memory updating is nonlinear, so the integral is not analytically tractable. A simple approximation is to use the mode of $q\left(\mathbf{w}_t\right)$, which is the mean $\mu_\mathbf{w}$. At this point, we carry forward the approximation of $q\left(\mathbf{M}_{t-1}\right)$ to $q\left(\mathbf{M}_t\right)$, which can be used for the $t+1$ step update. This procedure can start from $t = 0$ and continue until $t = T$; we thus obtain an approximate memory posterior $q\left(\mathbf{M}_T\right)$ by maximising the lower-bound $\mathcal{B}_T$. Thus, the sequential update of memory, as summarised in Algorithm 1, maximises a lower-bound of the episode log-likelihood.

## D  The Lease-Square Problem in Inference and Prediction

This sections shows that solving the same least squares problem is involved in both of the following problems:

1. minimising the KL-divergence between $\mathbf{w}$ during inference (section 3.1), $\mu_\mathbf{w}^* = \mathrm{argmin}_{\mu_\mathbf{w}}\mathrm{D_{KL}}\left(q\left(\mathbf{w}\right) \| p\left(\mathbf{w}|\mathbf{x}, \mathbf{M}\right)\right)$

2. approximating the predictive distribution $q(\hat{\mathbf{x}}|\mathbf{x}_q, \mathbf{M})$ (Section 3.3), $\mathbf{w}^* = \mathrm{argmax}_\mathbf{w}\, p(\mathbf{w}|\mathbf{x}_q, \mathbf{M})$

We first re-write the KL-divergence using its definition:

$$\mathrm{D_{KL}}\left(q\left(\mathbf{w}\right) \| p\left(\mathbf{w}|\mathbf{x}, \mathbf{M}\right)\right) = \int q\left(\mathbf{w}\right) \ln \frac{q\left(\mathbf{w}\right)}{p\left(\mathbf{w}|\mathbf{x}, \mathbf{M}\right)}\, \mathrm{d}\mathbf{w}$$
$$= -\mathrm{H}\left[q\left(\mathbf{w}\right)\right] - \langle \ln p\left(\mathbf{w}|\mathbf{x}, \mathbf{M}\right)\rangle_{q(\mathbf{w})} \tag{14}$$

where the first entropy term is a constant that depends on the fixed variance $\sigma_\mathbf{w}^2$. Therefore, minimising this KL-divergence is equivalent to maximising $\langle \ln p\left(\mathbf{w}|\mathbf{x}, \mathbf{M}\right)\rangle_{q(\mathbf{w})}$.

This posterior distribution over $\mathbf{w}$ can be expanded using Bayes' rule:

$$\ln p(\mathbf{w}|\mathbf{x}, \mathbf{M}) = \ln \frac{p\left(\mathbf{x}|\mathbf{w}, \mathbf{M}\right) p\left(\mathbf{w}\right)}{p(\mathbf{x}|\mathbf{M})}$$

$$= \ln p\left(\mathbf{x}|\mathbf{w}, \mathbf{M}\right) + \ln p\left(\mathbf{w}\right) + \dots \tag{15}$$

$$\approx -\frac{\left\|e(\mathbf{x}) - \mathbf{M}^{\mathsf{T}}\,\mathbf{w}\right\|^2}{2\sigma_\xi^2(\mathbf{x})} - \frac{1}{2}\left\|\mathbf{w}\right\|^2 + \dots$$

We omitted terms that do not depend on $\mathbf{w}$, including various normalising constants. In addition, the last line used the encoding projection $e(\mathbf{x}) \to \mathbf{z}$ to transform the distribution over $\mathbf{x}$ to that over $\mathbf{z}$. When $e(\mathbf{x})$ is invertible, the Jacobian factor $\frac{\mathbf{z}}{\partial \mathbf{x}} = \frac{\partial e(\mathbf{x})}{\partial \mathbf{x}}$ resulting from the distribution transform is well-defined and can be omitted since it does not depend on $\mathbf{w}$. However, the assumption of bijection is unlikely to be strictly satisfied by the neural network encoder/decoder pair, so the relation is approximate.

Taking the expectation of the above quadratic equation over the Gaussian distribution $q(\mathbf{w})$ results in the same quadratic form:

$$\left\langle \ln p\left(\mathbf{w}|\mathbf{x}, \mathbf{M}\right)\right\rangle_{q(\mathbf{w})} \approx \left\langle -\frac{\left\|e(\mathbf{x}) - \mathbf{M}^{\mathsf{T}}\,\mathbf{w}\right\|^2}{2\sigma_\xi^2} - \frac{1}{2}\left\|\mathbf{w}\right\|^2 \right\rangle_{q(\mathbf{w})} + \dots$$

$$= -\frac{\left\|e(\mathbf{x}) - \mathbf{M}^{\mathsf{T}}\,\mu_{\mathbf{w}}\right\|^2}{2\sigma_\xi^2} - \frac{1}{2}\left\|\mu_{\mathbf{w}}\right\|^2 + \frac{\sigma_{\mathbf{w}}^2}{2\sigma_\xi^2}\,\mathrm{Tr}\left(\mathbf{M}\,\mathbf{M}^{\mathsf{T}}\right) + \sigma_{\mathbf{w}}^2 C + \dots \tag{16}$$

where the last two terms do not depend on $\mu_{\mathbf{w}}$. Therefore, both inference and prediction involve solving the same least-squares problem.

## E   Proof of Attractor Dynamics

Here we show that in a well trained model, a pattern $\mathbf{x}^*$ in the memory is *asymptotically stable* under the dynamics, so that a state near $\mathbf{x}^*$ will converge to it. By "a well trained model", we assume that pattern $\mathbf{x}^*$ is a local maximum of the ELBO (eq. 4 in the main text)

$$\mathcal{L}(\mathbf{x}^*) = \left\langle \ln p\left(\mathbf{x}^*|\mathbf{w}^*, \mathbf{M}\right)\right\rangle_{q(\mathbf{M})} - \mathrm{D}_{\mathrm{KL}}\left(q(\mathbf{w}^*)\|p\left(\mathbf{w}\right)\right) - \mathrm{D}_{\mathrm{KL}}\left(q(\mathbf{M})\|p\left(\mathbf{M}|\mathbf{x}_{\leqslant T}\right)\right) \tag{17}$$

When $\mathcal{L}(\mathbf{x}^*)$ is at maximum, the energy we defined in eq. 14 (copied below) would be at its local minimum. This follows since the negative energy is just the first 2 terms of $\mathcal{L}$ without the KL-divergence between $\mathbf{M}$, which is a constant when the memory is fixed.

$$\mathcal{E}(\mathbf{x}, \mathbf{w}) = -\left\langle \ln p\left(\mathbf{x}|\mathbf{w}, \mathbf{M}\right)\right\rangle_{q(\mathbf{M})} + \mathrm{D}_{\mathrm{KL}}\left(q_t(\mathbf{w})\|p\left(\mathbf{w}\right)\right) \tag{18}$$

Section 3.5 of the main text shows $\mathcal{E}(\mathbf{x}_n, \mathbf{w}_n) \leqslant \mathcal{E}(\mathbf{x}_{n-1}, \mathbf{w}_{n-1})$ under the predictive dynamics. Therefore, we can construct a Lyapunov function candidate as:

$$V(\mathbf{x}, \mathbf{w}) = \mathcal{E}(\mathbf{x}, \mathbf{w}) - \mathcal{E}(\mathbf{w}^*, \mathbf{w}^*) \tag{19}$$

which satisfies:

$$V(\mathbf{x}^*, \mathbf{w}^*) = 0 \tag{20}$$

$$V(\mathbf{x}, \mathbf{w}) > 0 \quad \forall (\mathbf{x}, \mathbf{w}) \neq (\mathbf{x}^*, \mathbf{w}^*) \tag{21}$$

$$V(\mathbf{x}_n, \mathbf{w}_n) < V(\mathbf{x}_{n-1}, \mathbf{w}_{n-1}) \quad \forall (\mathbf{x}_n, \mathbf{w}_n) \neq (\mathbf{x}^*, \mathbf{w}^*) \tag{22}$$

Therefore, according to Lyapunov Stability theory, state $\mathbf{x}^*$ is asymptotically stable and serves as a point attractor in the system.

## F   Other Practical Considerations

For readers interested in applying the DKM, a few variants of Algorithm 1 may be worth considering. First, instead of using $\mathcal{L}_T$ in eq.8 (eq. 3 in the main text) as the objective, an alternative objective is $\mathcal{B}_T$. Although this lower-bound tends to be less tight than $\mathcal{L}_T$, it is also cheaper to compute. It may be particularly useful in online settings, since we only need to run through an episode once to compute $q\left(\mathbf{M}_T\right)$. This bounds can be further tightened by: 1. Using the optional step in Algorithm 1. We recommend starting with 2 or 3 steps. 2. Minimising a few other intermediate $\mathcal{B}_t$'s, for $0 < t < T$. This may be helpful in the case of long episodes wherein gradient propagation through the entire episode is infeasible.