[Reviews · NeurIPS 2018]

Reviewer 1



Authors propose an interesting insight into analyzing dynamics in CNNs. The paper is well written and straight forward to understand.

Reviewer 2



This paper proposes a generative model which builds on ideas from dynamical systems and previous deep learning work like the Kanerva Machine. The main idea is to design and train an architecture that, when unrolled as a dynamical system, has points from the target distribution as attractors. I found the presentation of the model reasonably clear, but thought it suffered from excessive formality. E.g., the description of p(M) could just say that the rows of M are isotropic Gaussian distributions with each row having its own mean and scaled-identity covariance. The references to matrix-variate Gaussians, Kronecker products, vectorization operators, etc. don't contribute to clarity. Similarly, talk about exchangeability doesn't add much. One could just say that the model treats all observations in an episode as independent, conditioned on the memory M. It's a matter of personal taste, but it seems to me that much of the technical presentation in the paper creeps into this sort of over-formalized style. The empirical results are difficult to evaluate, since the model kind of operates in its own domain, on tasks for which there aren't obviously useful benchmarks and baselines. Some quantitative results are presented, and the model seems to outperform the Kanerva Machine and Differentiable Neural Computer in terms of memory capacity. Qualitative results on a couple of denoising tasks are also presented, but it's not clear how we are supposed to interpret them. I think the general direction of this paper, i.e. generative models incorporating memory and aspects of dynamical systems, is interesting. But, I don't think the motivation provided by the presented experiments is strong enough for other researchers to go through the effort of reproducing this work. The paper could be improved by simplifying presentation of the technical content and by experiments exhibiting the benefits of the model in more natural tasks, e.g. as a component in an RL agent. --- I have read the author rebuttal.

Reviewer 3



This paper proposes a new paradigm for generative memory using attractor dynamics. This is an important advance in the generative memory models, such as the Hopfield network, Boltzmann machine, and Kanerva machine (KM). Specifically, this work presents a non-trivial dynamic version of the KM , and associated optimization (dynamic addressing & Bayesian memory update), and learning (training & prediction) rules. The new methods have been tested in two benchmark datasets, and compared with the result of KM. Overall, the conceptual idea is innovative, and technical foundation is sound. The result comparison is compelling. A couple of suggestions: 1) the memory M in the generative model is modeled as a Gaussian latent variable with a set of isotropic Gaussian distributions. If the columns of R are mutually independent, then V is diagonal , but U is generally a full matrix. Can that be generalized to an exponential family? 2) One of appealing feature of the dynamic KM is it is flat capacity curves (Fig. 2, right). I wonder what would be the capacity limit? is there any empirical rule to estimate the capacity? Does the capacity curve depend on the SNR? I would imagine a low SNR would induce a lower compression ratio. The denoising application shows noisy images with 15% salt-and-pepper noise, how does the performance change with the SNR or noise statistics? 3) The author shall discuss how to extend the idea from point attractors to more general cases such as line attractors. Also, what are the limitation of the dynamic KM? Finally, the authors shall consider sharing the open-source software to enhance the reproducibility.